# Exploring running styles in the field through cadence and duty factor modulation

**Anouk Nijs**[ID]*, **Melvyn Roerdink, Peter Jan Beek**[ID]

Department of Human Movement Sciences, Amsterdam Movement Sciences, Vrije Universiteit Amsterdam, Amsterdam, Netherlands

* a.nijs@vu.nl

**Data Availability Statement:** All relevant data are within the manuscript and its Supporting Information files.

**Funding:** This work was supported by the Dutch Research Council (NWO; https://www.nwo.nl/)

## Abstract

According to the dual-axis model, running styles can be defined by cadence and duty factor, variables that have been associated with running performance, economy and injury risk. To guide runners in exploring different running styles, effective instructions to modulate cadence and duty factor are needed. Such instructions have been established for treadmill running, but not for overground running, during which speed can be varied. In this study, five participants completed eight field training sessions over a 4-week training period with acoustic instructions to modulate cadence, duty factor, and, in combination, running style. Instructions were provided via audio files. Running data were collected with sports watches. Participants' experiences with guided-exploration training were evaluated with the user experience questionnaire. Data analysis revealed acoustic pacing and verbal instructions to be effective in respectively modulating cadence and duty factor, albeit with co-varying effects on speed and the non-targeted variable (i.e. duty factor or cadence). Combining acoustic pacing and verbal instructions mitigated these co-varying effects considerably, allowing for running-style modulations in intended directions (particularly towards the styles with increased cadence and increased duty factor). User experience of this form of guided-exploration training was overall positive, but could be improved in terms of autonomy (dependability). In conclusion, combining acoustic pacing and verbal instructions for running-style modulation is effective in overground running.

## Introduction

Running is a popular sport practiced by many people worldwide [1]. Plausible reasons for this are that running is an easily accessible type of physical activity in terms of preparation, location and cost [1], which is associated with significant health benefits [2]. Many runners use a sports watch or mobile application to monitor their running, but they seldom adapt their running style [3]. Runners can benefit from running-style modifications, especially if they are prone to injury or recently began running [4–6]. This suggests that incorporating effective instructions in existing mobile applications could help runners improve their running style in terms of injury prevention, running performance, or running economy, depending on the running variables being targeted.

under Grant P16–28 (Project 3). The funders had no role in study design, data collection and analysis, decision to publish, or preparation of the manuscript. There was no additional internal or external funding received for this study.

**Competing interests:** The authors have declared that no competing interests exist.

Cadence has been related to injury risk [7, 8] and running economy [6, 9, 10]. A common instruction method for modifying cadence is acoustic pacing. A metronome beat or music specifying the desired cadence is played, and the runner synchronizes their steps to the corresponding rhythm [11–13]. Acoustic pacing for cadence modulation was proven to be effective for running on a treadmill in the laboratory [11, 12]. An important difference between treadmill running and overground running is that changes in speed are restricted in treadmill running but not in overground running. In overground running, cadence is related to speed, with a higher speed corresponding to a higher cadence [14], although this is not a consistent finding [22]. Because of this, acoustic pacing to increase cadence could lead to a concomitant increase in speed during overground running. It is therefore important to examine the effectiveness of acoustic pacing in changing the cadence in overground running, while simultaneously considering speed changes. Some studies have used acoustic pacing in overground running on a track [13] or outdoors [15, 16] and found it to be effective. Participants in these studies were instructed to keep running speed constant, but possible changes in speed as a result of the cadence manipulation were often not analyzed specifically. Counterintuitively, te Brake and colleagues [15] reported an increase in cadence combined with a reduction in speed after a four-week music-based intervention aimed at increasing cadence.

Besides cadence, the duty factor (i.e. the ratio of stance time relative to step time) has been associated with injury risk [17], running economy, and performance [18]. Verbal instructions to change stance time and flight time were found to be effective in changing the duty factor when running at a constant speed on a treadmill [19–21]. To the best of our knowledge, instructions to change the duty factor have not been investigated in overground running to date. The duty factor is also associated with speed in that a higher speed corresponds to a lower duty factor (i.e. a shorter stance time relative to the step time [14]). Hence, similar to cadence, instructions to change the duty factor could elicit concomitant variations in speed. Furthermore, as cadence and duty factor are both associated with step time, instructions to change either variable could affect the other variable as well. A change in cadence without a change in duty factor requires a change in both stance time and flight time without changing the ratio between them. When studying the effects of modulating a specific running variable, it is therefore important not only to quantify the effects for that specific variable but also to quantify potential co-varying effects on other running variables, especially in studies on running style modulation. Such co-varying effects have not been reported in the literature for modulated overground running, while for constant-speed treadmill running instructions aimed at changing the duty factor did not affect cadence [19].

According to the dual-axis model [22], running styles can be categorized at a certain speed through the combination of cadence and duty factor (Fig 1). The basic idea behind the model is that these two ('distal') variables respectively reflect the ('proximal') horizontal and vertical displacement of the center of mass, resulting from the interplay of many kinematic and kinetic factors [22]. The dual-axis model distinguishes five different running styles. The 'Sit', located at the center of the model, represents the average runner, with an average cadence and duty factor. The 'Hop', located on the left of the model, represents a high cadence, and thus a short step, leading to a small horizontal displacement per step. The 'Push' on the contrary, located on the right side, reflects a low cadence, and thus a long step and large horizontal displacement per step. The 'Bounce', located at the top of the model, reflects a low duty factor, and thus a relatively long flight phase, corresponding to a larger vertical displacement. Finally, the 'Stick', on the bottom, reflects a high duty factor, and thus a relatively long stance time, corresponding to a lower vertical displacement. Thus, according to the dual-axis model, modulating cadence or duty factor in a specific direction allows for modulating one's running style. In this study, we combined acoustic pacing to modulate cadence and verbal instruction to modulate duty factor

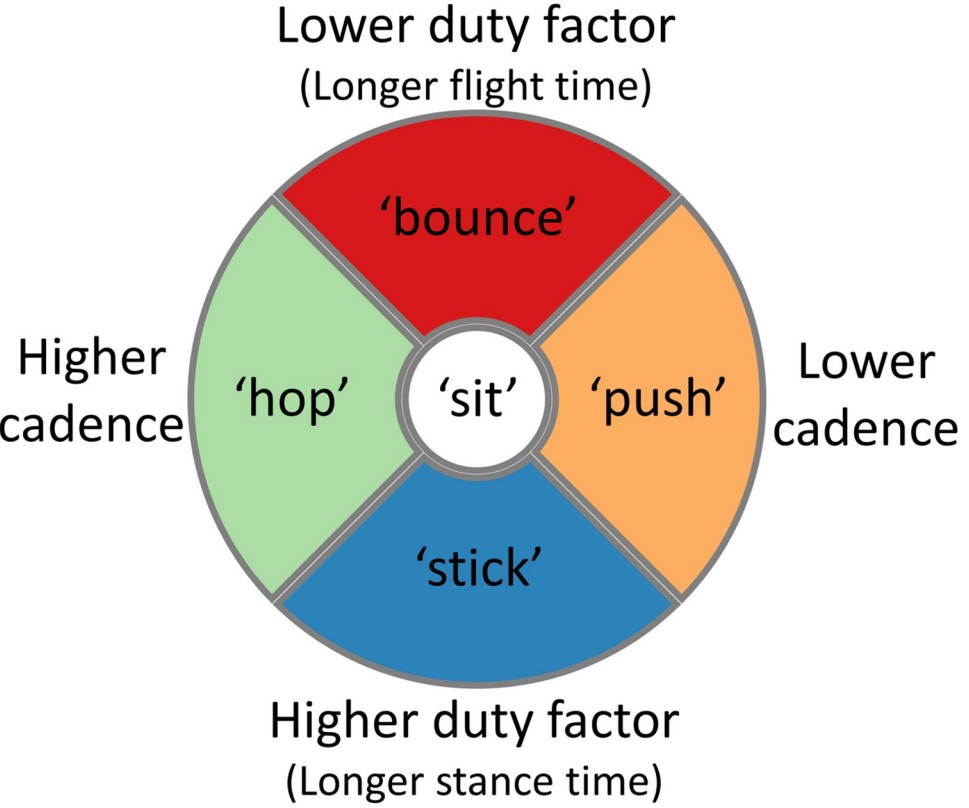

**Fig 1. Visual representation of the dual-axis model.**

in order to modulate running style (e.g. by combining pacing at the preferred cadence with a verbal instruction to decrease stance time, thereby guiding the runner in the direction of the 'Bounce' style).

This study had three aims. 1) To examine the effects of the acoustic pacing and verbal instructions regarding stance time on cadence, duty factor, and speed. In this regard we focused on both the targeted effects and potential co-varying effects of the instructions on speed and on the non-targeted variable (i.e. duty factor with acoustic pacing and cadence with verbal instructions). We hypothesized that in overground running both acoustic pacing and verbal instructions result in the targeted effects, but to a lesser extent than on a treadmill, due to co-varying effects on running speed. 2) To examine the effectiveness of combined acoustic-pacing and verbal-instruction conditions in guiding participants towards a certain running style, as defined by the dual-axis model (Fig 1). Also in this context, possible effects on speed were considered. We expected the combined instructions to be more effective than the individual instructions, due to a smaller chance of co-varying effects of acoustic pacing on the duty factor and verbal instructions regarding stance time on the cadence. 3) We aimed to assess the user experience of a 4-week guided-exploration training program with these acoustic instructions in the field.

## Materials and methods

For this study a convenience sample of five healthy adult recreational runners (Table 1) was recruited. All participants had multiple years of running experience (Table 1), and ran multiple times per week (Table 1). They routinely used a sports watch with an accessory (e.g., heart rate

**Table 1. Participants' age, years of running experience, training frequency, determined baseline speed, number of runs at baseline speed, and baseline cadence, stance time, and duty factor.**

| | Age (years) | Sex (male /female) | Experience (years) | Training frequency (training /week) | Number of valid runs | Baseline speed (km/h) | Number of runs at baseline speed | Baseline cadence (steps /min) | Baseline stance time (ms) | Baseline duty factor |
|---|---|---|---|---|---|---|---|---|---|---|
| 1 | 19 | f | 3 | 2 | 11 | 11 | 5 | 182 | 260 | 0.39 |
| 2 | 56 | m | 15 | 3 | 154 | 11 | 75 | 170 | 260 | 0.37 |
| 3 | 60 | m | 40 | 4 | 209 | 10 | 123 | 166 | 260 | 0.36 |
| 4 | 50 | m | 20 | 2.5 | 144 | 11 | 65 | 182 | 250 | 0.38 |
| 5 | 49 | m | 9 | 4 | 30 | 9 | 13 | 158 | 325 | 0.43 |

belt or footpod) measuring cadence and stance time, as required for the present purpose to assess the effects of pacing and instructions. The recruited number of participants was relatively small due to this requirement. Participants were recruited via social networks and athletics clubs. Recruitment was stopped when after 6 months of active recruitment (between December 2021 and May 2022), only five applicants met the requirements. In view of this small sample of suitable volunteers, we opted for nonparametric statistical testing within each individual participant to best answer the research questions. All participants provided written informed consent before the start of the study. The study protocol was in compliance with the Declaration of Helsinki and approved by the Scientific and Ethical Review Board (VCWE) of the Faculty of Behavioural and Movement Sciences of the Vrije Universiteit Amsterdam. All participants were given a participant number. Only author AN had access to the key for this pseudonymization.

To personalize acoustic pacing and verbal instructions, baseline values for speed, cadence and duty factor were required. Therefore, participants shared their running data collected on their personal sports watch for at least the month before the training period. This data was shared through the online platform of Move-Metrics (Ede, the Netherlands), and parameterized per training in terms of date, duration, and distance, and mean, standard deviation, median, inter quartile range (IQR), minimum, maximum, and lower (5%) and upper (95%) limit of the confidence interval for speed, cadence, and stance time while running. Based on these data, the baseline speed was determined as the rounded median speed over the training sessions. The median instead of the mean was used to reduce the effect of possible outliers when for example a training was labelled incorrectly. The baseline cadence and stance time were determined as the median cadence and stance time over the training sessions of which the rounded median speed was at the baseline speed. Duty factor was calculated based on the stance time and cadence according to:

$$duty\ factor = \frac{stance\ time}{(60/cadence)*2},\qquad(\text{Eq1})$$

where stance time is expressed in seconds and (60/cadence) represents the step time in seconds, rendering the duty factor a dimensionless variable (Table 1).

An audio file (see S1 Audio for an example) was created with verbal instructions to keep speed constant at the baseline speed and explore cadence and stance time relative to the baseline values guided by acoustic pacing and verbal instructions regarding stance time. Participants trained according to the instructions twice a week for a period of four weeks by playing the audio file on a device of their choice while running. They were instructed to start the audio file and the measurement on their sports watch simultaneously. Short walking blocks were

included at specific times in the audio file, allowing synchronization of the training data to the audio file instructions.

For the resultant eight training sessions, we received speed, cadence and stance time for each second of training. The data during the eight training sessions were normalized relative to the set target values. We then calculated the median speed, cadence and duty factor for each instruction.

## Cadence and duty-factor modulation

To assess the effects of the individual instructions, we calculated the slope of the change in cadence, duty factor and speed as a result of the increase in pacing frequency for each training. For each participant, we then calculated the 95% confidence interval and used a one-sample Wilcoxon signed rank test over the eight training sessions to examine if the slopes were significantly different from zero ($p < 0.05$), which would indicate an effect of the acoustic pacing. Effect size $r$ was calculated as $r = |Z/\sqrt{N}|$, where $Z$ is the standardized test statistic and $N$ is the number of training sessions. Cadence, duty factor and speed were also compared between the instruction to increase stance time and the instruction to decrease stance time using a Wilcoxon signed rank test to assess the effect of the verbal instructions to change stance time.

## Running-style modulation

Since the dual-axis model has been introduced only recently, no population reference values for the axes are yet available. We therefore decided to modulate running style relative to participant's baseline running style and regarded that as the 'Sit' style in the center of the model. By giving combined acoustic pacing and verbal stance-time instructions we then aimed to guide participants away from their baseline running style towards one of the four other running styles (Table 2). The same audio file was used in all eight training sessions, and the order in which the instructions were given was the same for all participants.

We defined cadence, duty factor and speed for each participant's 'Sit' baseline running style, and compared cadence, duty factor and speed observed for each of the other four modulated running styles to these baseline values using Wilcoxon signed rank tests. The 95% confidence intervals for the mean differences were also calculated.

## Subjective user experience

After the four-week training period, participants filled out a questionnaire on their experience with the guided-exploration training. For this purpose, the Dutch version of the User Experience Questionnaire (UEQ) was used ([23]; translated by Adriaan Dekker according to [24]; obtained from www.ueq-online.org). The UEQ consist of 26 pairs of opposing terms on the

**Table 2. Overview of the instructions given to explore each running style.**

| Instruction | | |
|---|---|---|
| **Speed** | **Cadence** | **Stance time** |
| Run at the constant baseline speed | 1.00 * baseline cadence | Increase stance time ('Stick') |
| | 1.10 * baseline cadence ('Hop') | Stop increasing stance time |
| | 1.00 * baseline cadence (Baseline running style) | |
| | 0.90 * baseline cadence ('Push') | |
| | 1.00 * baseline cadence | Decrease stance time ('Bounce') |

extremes of a 7-point Likert scale. Answers range from -3 (completely agree with the term on the left side) to +3 (completely agree with the term on the right side). Each of the 26 items belongs to one of six scales: (1) Attractiveness, what is the user's overall impression of the product?; (2) Perspicuity, is it easy to understand?; (3) Efficiency, is the interaction deemed efficient?; (4) Dependability, does the user feel in control and safe?; (5) Stimulation, is it exciting to use the product?; and (6) Novelty, does it capture the users' attention? For each individual, the UEQ scores were grouped according to the six scales of the UEQ. These scores were then compared to the established benchmark values for the UEQ [23].

## Results

### Cadence and duty factor modulation

The slope for cadence over the increasing acoustic pacing frequencies was positive and significantly different from zero for all participants, indicating that they were able to change cadence when modulated by acoustic pacing (Fig 2A; Table 3). For the duty factor, acoustic pacing did not have a consistent systematic effect. The slope was only significant for participants 2 (positive) and 3 (negative), indicating an increase and decrease in duty factor with increasing acoustic pacing frequency, respectively (Fig 2C; Table 3). The slope for speed was positive and significant for all participants, indicating an increase in speed with increasing pacing frequency (Fig 2E; Table 3).

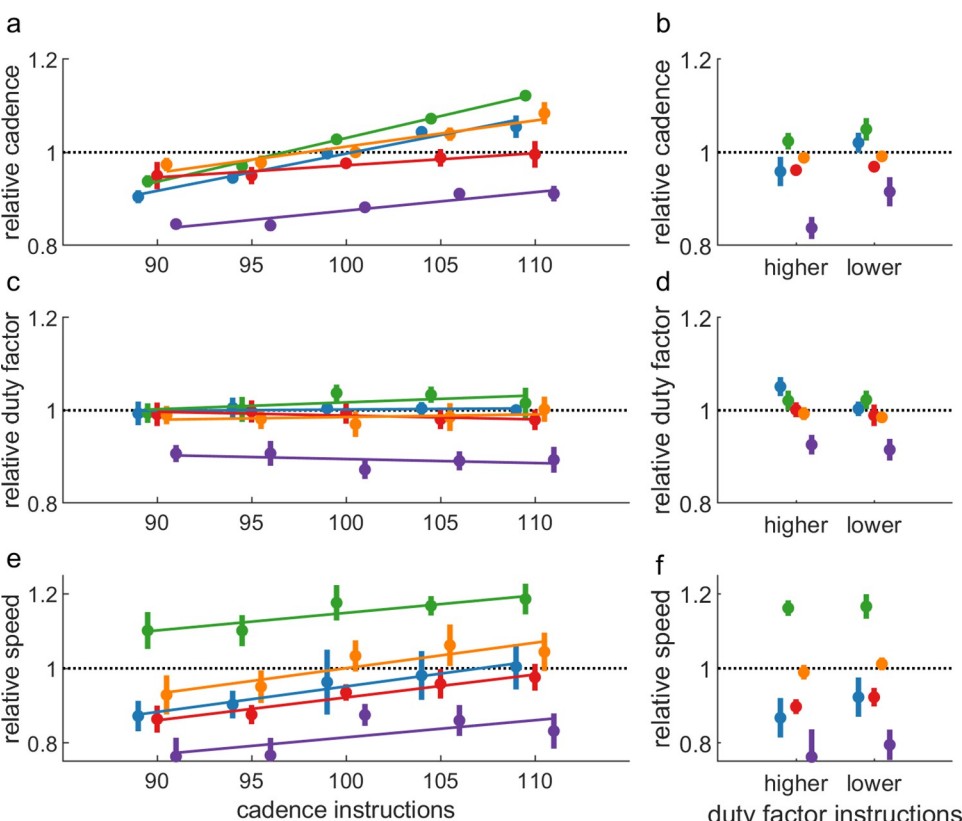

**Fig 2. Mean cadence, duty factor, and speed over the eight training sessions, relative to baseline values.** The left panels (a, c, and e) show the values and regression lines for cadence modulation with acoustic pacing. The right panels (b, d, and f) show the values for the two verbal instructions to change the duty factor. The vertical lines represent the standard deviation between the eight training sessions.

**Table 3. Mean slope of the targeted variable cadence and the two potentially co-varying variables duty factor and speed as a function of acoustically paced cadence modulations (center rows) and mean difference between the verbal instructions to increase and decrease stance time (increase–decrease) in the targeted variable duty factor (positive values indicate a change in the modulated direction) and potentially co-varying variables cadence and speed (lower rows).** Slope and difference data are complemented by 95% confidence intervals and the results of the Wilcoxon Signed Rank test. Expected changes for targeted (T) and potentially co-varying (C) variables are shown at the top of each block; results in line with these expectations are highlighted in green.

| modulated variable | | Cadence | | | | Duty factor | | | | Speed | | | |
|---|---|---|---|---|---|---|---|---|---|---|---|---|---|
| **Cadence** | expectation | Positive slope (T) | | | | no slope (C) | | | | no slope (C) | | | |
| | Participant | slope | 95% confidence | | Sig. | r | slope | 95% confidence | | Sig. | r | slope | 95% confidence | | Sig. | r |
| | 1 | 0.799 | 0.696 | 0.903 | 0.012* | 0.893 | 0.025 | -0.077 | 0.127 | 0.401 | 0.297 | 0.686 | 0.402 | 0.970 | 0.012* | 0.891 |
| | 2 | 0.940 | 0.895 | 0.984 | 0.012* | 0.893 | 0.149 | 0.018 | 0.280 | 0.050* | 0.693 | 0.471 | 0.322 | 0.620 | 0.012* | 0.891 |
| | 3 | 0.259 | 0.028 | 0.490 | 0.012* | 0.893 | -0.082 | -0.136 | -0.027 | 0.025* | 0.792 | 0.615 | 0.345 | 0.884 | 0.012* | 0.891 |
| | 4 | 0.562 | 0.422 | 0.703 | 0.012* | 0.893 | 0.058 | -0.006 | 0.122 | 0.093 | 0.594 | 0.685 | 0.412 | 0.957 | 0.012* | 0.891 |
| | 5 | 0.453 | 0.348 | 0.557 | 0.012* | 0.893 | -0.098 | -0.222 | 0.026 | 0.327 | 0.346 | 0.563 | 0.209 | 0.916 | 0.012* | 0.891 |
| **Duty factor** | expectation | no difference (C) | | | | increase (T) | | | | no difference (C) | | | |
| | Participant | difference | 95% confidence | | Sig. | r | difference | 95% confidence | | Sig. | r | difference | 95% confidence | | Sig. | r |
| | 1 | -0.062 | -0.085 | -0.039 | 0.012* | 0.893 | 0.048 | 0.029 | 0.067 | 0.012* | 0.891 | -0.055 | -0.099 | -0.012 | 0.012* | 0.891 |
| | 2 | -0.026 | -0.041 | -0.010 | 0.012* | 0.893 | 0.053 | 0.032 | 0.073 | 0.012* | 0.891 | -0.004 | -0.036 | 0.027 | 0.575 | 0.198 |
| | 3 | -0.007 | -0.015 | -0.000 | 0.066 | 0.651 | 0.051 | 0.031 | 0.072 | 0.012* | 0.891 | -0.025 | -0.045 | -0.006 | 0.025* | 0.792 |
| | 4 | -0.003 | -0.010 | 0.004 | 0.340 | 0.337 | 0.050 | 0.030 | 0.070 | 0.012* | 0.891 | -0.022 | -0.034 | -0.011 | 0.012* | 0.891 |
| | 5 | -0.089 | -0.125 | -0.052 | 0.011* | 0.897 | 0.044 | 0.027 | 0.062 | 0.012* | 0.891 | -0.040 | -0.094 | 0.015 | 0.161 | 0.495 |

Duty factor was significantly different between the two verbal stance-time instructions for all participants, varying in the instructed directions (Fig 2D; Table 3). Cadence was significantly different between the instructions for participants 1, 2, and 5 and speed was significantly different for participants 1, 3, and 4, with a higher cadence and speed after the instruction to decrease stance time (aimed at a lower duty factor; Fig 2B and 2F; Table 3).

## Running-style modulation

Manipulations towards the 'Stick' running style (increasing duty factor with verbal instructions to increase stance time) led to a significant increase in the targeted duty factor compared to baseline for all participants (Fig 3B; Table 4). Co-varying effects were minimal: cadence was significantly higher for participants 1 and 4 while speed was significantly higher for participant 2 only (Fig 3A and 3C; Table 4).

Manipulations towards the 'Hop' running style (increasing cadence with faster acoustic pacing) led to a significant increase in the targeted cadence compared to baseline for participants 1, 2, 4, and 5 (Fig 3A; Table 4). Again, co-varying effects were limited: duty factor did not differ significantly from baseline for any of the participants, while speed only increased significantly for participants 4 and 5 (Fig 3B and 3C; Table 4).

Manipulations towards the 'Push' running style (decreasing cadence with slower acoustic pacing) led to a significant decrease in the targeted cadence for all participants (Fig 3A; Table 4). Co-varying effects were more pronounced: duty factor decreased significantly for all participants and speed decreased significantly for participants 3 and 4 (Fig 3B and 3C; Table 4).

Manipulations towards the 'Bounce' running style (decreasing duty factor with verbal instructions to decrease stance time) did not lead to a significant reduction in the duty factor compared to baseline in any of the participants (Fig 3B; Table 4). In the absence of an effect on the targeted variable, also co-varying effects were largely absent: only the speed of participant 4 was significantly higher compared to baseline (Fig 3A and 3C; Table 4).

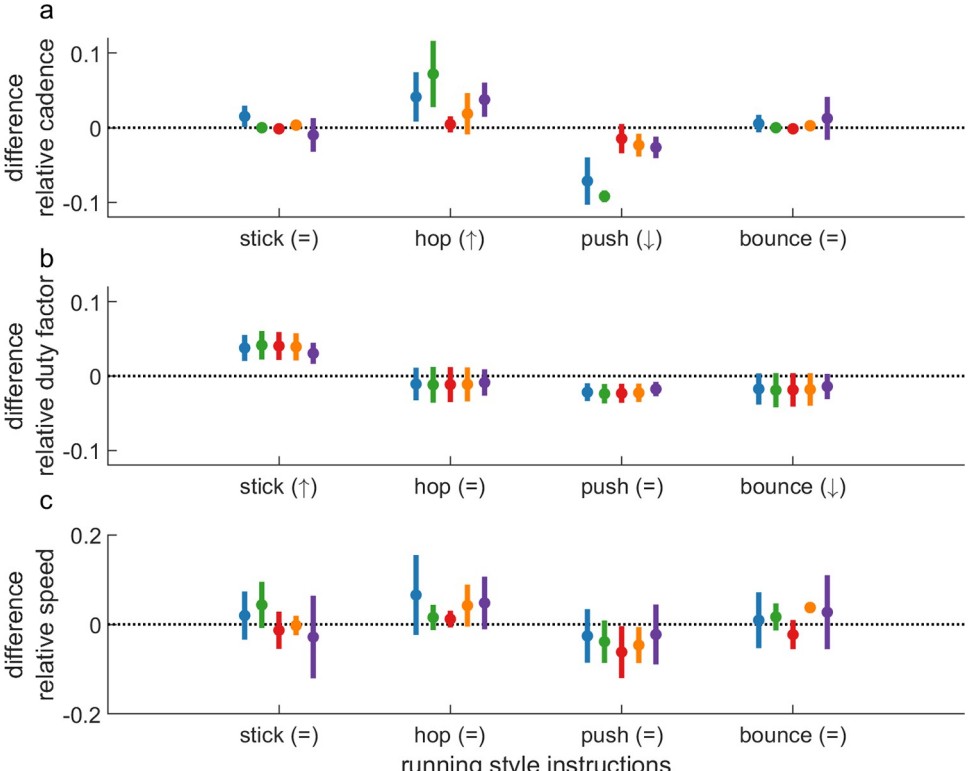

**Fig 3.** Difference in the relative cadence (a), duty factor (b), and speed (c) over the eight training sessions under the combined manipulation of acoustic pacing and verbal stance-time instructions to modulate participant's baseline running style towards 'Stick', 'Hop', 'Push' and 'Bounce' running styles. The symbols between brackets indicate the targeted change, where the arrows up and down respectively represent a targeted increase and decrease, and the equal sign represents the targeted absence of a change. The vertical lines represent the standard deviation over the eight training sessions. The horizontal dotted line represents participant's baseline running style.

## Subjective user experience

The scores for the six scales of the UEQ are presented in Fig 4 for all participants. The established benchmark scores for the different scales are in the background of the figure [23]. As can be seen in the figure, the scores were on average excellent for Perspicuity, good for Novelty, on the border between above average and good for Stimulation, above average for Attractiveness, Efficiency, and bad for Dependability. The participants rated the guided-exploration training differently, as shown by the relatively large range in the ratings between participants on most scales.

## Discussion

In this study, we examined the effects of acoustic pacing for cadence modulation and verbal instruction for duty-factor modulation on cadence, duty factor and speed in overground running in the open field, as well as the effect of running-style modulation towards 'Stick', 'Hop', 'Push', and 'Bounce' running styles, relative to the participant's baseline running style, interpreted for the sake of the study as the 'Sit' running style. In addition, we examined the subjective evaluation of these instructions aimed at exploring different running styles during training.

We expected acoustic pacing to be effective at modulating the cadence and verbal stance-time instructions to be effective at changing the duty factor, although to a lesser extent than on

**Table 4. Mean change in cadence, duty factor and speed as a result of the combined manipulations of acoustic pacing and verbal stance-time instructions to modulate participant's baseline running style towards 'Stick', 'Hop', 'Push' and 'Bounce' running styles, as well as 95% confidence intervals and the results of the Wilcoxon Signed Rank test.** Expected changes for targeted (T) and potentially co-varying (C) variables are shown at the top of each running style block; results in line with the manipulations are highlighted in green.

| Running style | | Cadence | | | | | Duty factor | | | | | Speed | | | | |
|---|---|---|---|---|---|---|---|---|---|---|---|---|---|---|---|---|
| 'Stick' | expectation | no change (C) | | | | | Increase (T) | | | | | no change (C) | | | | |
| | Participant | mean | 95% confidence | | Sig. | r | mean | 95% confidence | | Sig. | r | mean | 95% confidence | | Sig. | r |
| | 1 | 0.015 | 0.003 | 0.027 | 0.026* | 0.789 | 0.038 | 0.023 | 0.052 | 0.012* | 0.891 | 0.020 | -0.025 | 0.065 | 0.401 | 0.297 |
| | 2 | 0.000 | 0.000 | 0.000 | 1.000 | 0.000 | 0.041 | 0.025 | 0.057 | 0.012* | 0.891 | 0.043 | 0.000 | 0.087 | 0.036* | 0.742 |
| | 3 | -0.001 | -0.005 | 0.002 | 0.317 | 0.354 | 0.040 | 0.025 | 0.056 | 0.012* | 0.891 | -0.013 | -0.048 | 0.022 | 0.528 | 0.223 |
| | 4 | 0.003 | 0.001 | 0.006 | 0.042* | 0.718 | 0.039 | 0.024 | 0.054 | 0.012* | 0.891 | -0.003 | -0.021 | 0.016 | 0.779 | 0.099 |
| | 5 | -0.011 | -0.033 | 0.010 | 0.276 | 0.385 | 0.035 | 0.021 | 0.048 | 0.012* | 0.891 | -0.034 | -0.129 | 0.060 | 0.401 | 0.297 |
| 'Hop' | expectation | Increase (T) | | | | | no change (C) | | | | | no change (C) | | | | |
| | Participant | mean | 95% confidence | | Sig. | r | mean | 95% confidence | | Sig. | r | mean | 95% confidence | | Sig. | r |
| | 1 | 0.041 | 0.014 | 0.069 | 0.011* | 0.897 | -0.011 | -0.029 | 0.008 | 0.093 | 0.594 | 0.066 | -0.009 | 0.140 | 0.069 | 0.643 |
| | 2 | 0.072 | 0.035 | 0.109 | 0.020* | 0.825 | -0.012 | -0.032 | 0.008 | 0.093 | 0.594 | 0.016 | -0.008 | 0.039 | 0.208 | 0.445 |
| | 3 | 0.004 | -0.005 | 0.013 | 0.257 | 0.401 | -0.011 | -0.031 | 0.008 | 0.093 | 0.594 | 0.012 | -0.004 | 0.027 | 0.123 | 0.545 |
| | 4 | 0.019 | -0.004 | 0.042 | 0.027* | 0.780 | -0.011 | -0.030 | 0.008 | 0.093 | 0.594 | 0.046 | 0.003 | 0.081 | 0.036* | 0.742 |
| | 5 | 0.043 | 0.021 | 0.065 | 0.011* | 0.897 | -0.010 | -0.027 | 0.007 | 0.093 | 0.594 | 0.058 | -0.002 | 0.118 | 0.050* | 0.693 |
| 'Push' | expectation | Decrease (T) | | | | | no change (C) | | | | | no change (C) | | | | |
| | Participant | mean | 95% confidence | | Sig. | r | mean | 95% confidence | | Sig. | r | mean | 95% confidence | | Sig. | r |
| | 1 | -0.071 | -0.098 | -0.045 | 0.011* | 0.896 | -0.022 | -0.032 | -0.012 | 0.012* | 0.891 | -0.026 | -0.076 | 0.024 | 0.263 | 0.396 |
| | 2 | -0.092 | -0.098 | -0.085 | 0.011* | 0.897 | -0.024 | -0.035 | -0.013 | 0.012* | 0.891 | -0.039 | -0.078 | 0.001 | 0.080 | 0.619 |
| | 3 | -0.015 | -0.031 | 0.002 | 0.039* | 0.728 | -0.023 | -0.034 | -0.013 | 0.012* | 0.891 | -0.062 | -0.110 | -0.013 | 0.017* | 0.843 |
| | 4 | -0.023 | -0.036 | -0.011 | 0.012* | 0.892 | -0.023 | -0.033 | -0.012 | 0.012* | 0.891 | -0.046 | -0.080 | -0.013 | 0.025* | 0.792 |
| | 5 | -0.030 | -0.044 | -0.016 | 0.011* | 0.898 | -0.020 | -0.029 | -0.011 | 0.012* | 0.891 | -0.028 | -0.096 | 0.041 | 0.441 | 0.273 |
| 'Bounce' | expectation | no change (C) | | | | | Decrease (T) | | | | | no change (C) | | | | |
| | Participant | mean | 95% confidence | | Sig. | r | mean | 95% confidence | | Sig. | r | mean | 95% confidence | | Sig. | r |
| | 1 | 0.005 | -0.004 | 0.015 | 0.180 | 0.474 | -0.017 | -0.035 | 0.000 | 0.069 | 0.643 | 0.009 | -0.043 | 0.061 | 0.889 | 0.049 |
| | 2 | 0.000 | 0.000 | 0.000 | 1.000 | 0.000 | -0.019 | -0.038 | 0.000 | 0.069 | 0.643 | 0.017 | -0.009 | 0.042 | 0.093 | 0.594 |
| | 3 | -0.001 | -0.005 | 0.002 | 0.317 | 0.354 | -0.019 | -0.037 | 0.000 | 0.069 | 0.643 | -0.023 | -0.050 | 0.004 | 0.092 | 0.595 |
| | 4 | 0.003 | -0.003 | 0.008 | 0.271 | 0.389 | -0.018 | -0.036 | 0.000 | 0.069 | 0.643 | 0.038 | 0.027 | 0.048 | 0.012* | 0.891 |
| | 5 | 0.014 | -0.013 | 0.042 | 0.236 | 0.419 | -0.016 | -0.032 | 0.000 | 0.069 | 0.643 | 0.033 | -0.051 | 0.118 | 0.528 | 0.223 |

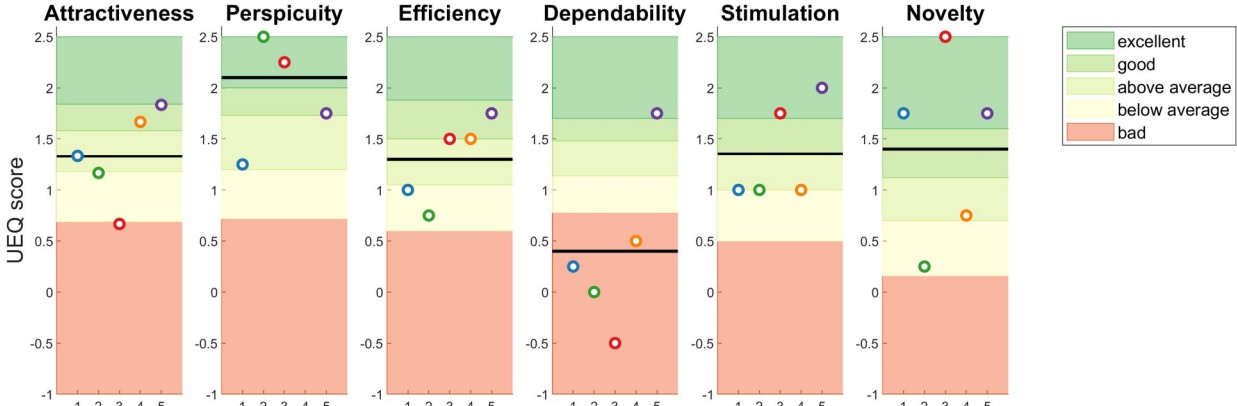

**Fig 4. The scores on the different user experience scales for all participants.** The horizontal black line is the mean over the participants. Shaded areas represent the benchmark values of <25% (bad), 25%-50%, 50%-75%, 75%-90%, and >90% (excellent) performance.

a treadmill [19] in view of the possibility to adapt the speed while running overground. Our results confirmed the expectation for acoustic pacing, as cadence did indeed increase with increasing pacing frequencies. Speed also increased with increasing pacing frequencies. Compared to a treadmill study, in which cadence relative to baseline cadence was 0.91, 1.00, and 1.11 in response to acoustic pacing at 90%, 100%, and 110% of baseline cadence [11], corresponding to a slope around 1.00, the change was indeed smaller for most participants in this study. Only participant 2 came close to this, with a mean slope of 0.94 (Table 3). For the duty factor, instructions to increase and decrease stance time led to changes in the targeted duty factor, but also to changes in speed and cadence. Compared to another treadmill study [19], in which duty-factor changes of about 10% relative to baseline were found (i.e. a difference between instructions of around 20%), the instructions were considerably less effective, with differences between duty-factor instructions of around 5%. These relatively small changes in duty factor could be due to the simultaneous changes in cadence and speed (Table 3). Overall, we were able to modulate the targeted variables with acoustic pacing or stance-time instructions in the right directions, but with lower magnitude of effect compared to fixed-speed treadmill conditions given the observed co-varying effects on the non-targeted variables. If these instructions are used in practice, it is important to keep the covarying effects in mind.

We expected the combined use of acoustic pacing and stance-time instructions to have a lower chance of co-varying effects of acoustic pacing on the duty factor and verbal stance-time instructions on the cadence, and thus to be more effective. In terms of the magnitude of the changes in the targeted variables of the running-style modulation, the combined duty-factor modulation was slightly more effective ('Stick': |0.0386 | + 'Bounce': |-0.0178| = 0.056) compared to verbal stance-time instructions alone (0.049). The combined cadence modulation was slightly less effective ('Hop': 0.036 and 'Push': -0.046) compared to acoustic pacing alone (110%: 0.054 and 90%: -0.059). More importantly, in line with our expectation, co-varying effects were smaller, with fewer changes in cadence as a result of the instructions towards the 'Stick' and 'Bounce' (which required a change in duty factor). Likewise, co-varying effects on duty factor were smaller for the 'Hop' (which required an increase in cadence), but not for the 'Push' (which required a decrease in cadence): duty factor decreased significantly for 'Push' running-style modulation for all participants. We further expected a co-varying effect on speed as a result of the acoustic-pacing variations to modulate running styles towards 'Hop' and 'Push' in a similar manner as for the acoustic-pacing conditions alone. However, this was not the case; in fact, there were fewer significant changes in speed for all running styles compared to the acoustic-pacing conditions alone (Tables 3 vs 4). Overall, the combined instructions for running-style modulation were effective at changing the targeted variables, with fewer and smaller co-varying changes in the non-targeted variables, most notably speed. This indicates that it is better to use the combined instructions in practice.

In general, user experience of the guided-exploration training was positive for all domains, except for the scale Dependability. Dependability reflects the extent to which the user feels in control and safe [23]. In this study, an audio file was used to administer the instructions during the training. As a result, the instructions were exactly the same for every training, and the users had no control over the instructions at all, which could explain the lower scores on this scale. If the guided-exploration training tested in this study was to be implemented in a mobile application, where the user can change settings and have more autonomous control over which instructions are provided, we expect this score to go up. In that situation, the already higher experience scores on the other scales might follow suit.

We defined the participants' preferred running style as the baseline running style and induced and evaluated any changes relative to this running style. However, the preferred running style of a participant is not necessarily the 'Sit' running style according to the dual-axis

model and hence could influence their ability to change in certain directions, because there is a limit to these variables (e.g. when someone is a pronounced 'Bounce' runner (a relatively long flight phase), it could be more difficult to decrease duty factor even further, as the relative flight phase cannot be increased indefinitely). Unfortunately, there are currently no population reference values available for the different running styles on the basis of which the running style of an individual runner could be determined. For this, further research and speed-dependent reference values for different running styles are necessary.

In this study only five runners participated, mainly due to the limiting requirement of using a sports watch and accessory that measured stance time. As stance time and duty factor have only recently been recognized as important running variables [17, 22], not many wearables or mobile apps currently can measure and report these variables. For this study, the measurement of this variable was necessary to assess the effect of the instructions on the duty factor. However, this is not a requirement for the use of the instructions per se for a guided exploration towards different running styles, as one would only need a device to play the audio file.

Furthermore, participants used their own sports watch to measure the data. While sports watches are used in research and generally seem to provide valid measurements [9, 15, 16, 25], it should be noted that different devices were used and the accuracy of the data could depend on the specific software and hardware and could vary between participants. Because the data was analysed per participant, effects of low validity should be limited, but low reliability could have affected the results.

In conclusion, our results show that when acoustic pacing or verbal stance-time instructions are provided in overground running, the targeted variable (cadence or duty factor, respectively) can be successfully modulated, but with co-varying effects on the non-targeted variables (speed and respectively duty factor or cadence) due to mutual dependencies among these variables. Combining acoustic pacing and verbal stance-time instructions for running-style modulation largely mitigated the number and magnitude of co-varying effects. Our results indicate that these combined instructions are especially effective when increasing the cadence (modulation towards a more 'Hop'-like running style) and increasing the duty factor (modulation towards a more 'Stick'-like running style). Overall, users were generally positive about the 4-week guided-exploration training, except for the degree of Dependability. We expect that increasing the autonomy of the user by implementing the instructions in an application with self-control options will help enhance user experience of guided exploration of running styles.

## Supporting information

**S1 Dataset.**
(CSV)

**S1 Audio.**
(MP3)

## Acknowledgments

We would like to thank Ben van Oeveren and Move-Metrics for their technical support.

This study was part of a larger consortium project with Dopple B.V. (Assen, The Netherlands) and foundational for the joint development of the Running Buddy application for guided exploration of running styles using their instrumented earbuds to measure and modify running parameters.

## Author Contributions

**Conceptualization:** Anouk Nijs, Melvyn Roerdink, Peter Jan Beek.

**Data curation:** Anouk Nijs.

**Formal analysis:** Anouk Nijs.

**Funding acquisition:** Peter Jan Beek.

**Investigation:** Anouk Nijs.

**Methodology:** Anouk Nijs, Melvyn Roerdink, Peter Jan Beek.

**Project administration:** Anouk Nijs, Peter Jan Beek.

**Resources:** Peter Jan Beek.

**Software:** Anouk Nijs.

**Supervision:** Melvyn Roerdink, Peter Jan Beek.

**Validation:** Anouk Nijs, Melvyn Roerdink.

**Visualization:** Anouk Nijs, Melvyn Roerdink.

**Writing – original draft:** Anouk Nijs.

**Writing – review & editing:** Anouk Nijs, Melvyn Roerdink, Peter Jan Beek.

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
