## [Decision Letter · Decision Letter 0]

15 Aug 2023

PONE-D-23-18858Exploring running styles in the field through cadence and duty factor modulationPLOS ONE

Dear Dr. Nijs,

Thank you for submitting your manuscript to PLOS ONE. After careful consideration, we feel that it has merit but does not fully meet PLOS ONE’s publication criteria as it currently stands. Therefore, we invite you to submit a revised version of the manuscript that addresses the points raised during the review process, that pointed weaknesses on both the substance and the forme. Please submit your revised manuscript by Sep 29 2023 11:59PM. If you will need more time than this to complete your revisions, please reply to this message or contact the journal office at plosone@plos.org. Please include the following items when submitting your revised manuscript:A rebuttal letter that responds to each point raised by the academic editor and reviewer(s). You should upload this letter as a separate file labeled 'Response to Reviewers'.A marked-up copy of your manuscript that highlights changes made to the original version. You should upload this as a separate file labeled 'Revised Manuscript with Track Changes'.An unmarked version of your revised paper without tracked changes. You should upload this as a separate file labeled 'Manuscript'.

We look forward to receiving your revised manuscript.

Kind regards,

Laurent Mourot

Section Editor

PLOS ONE

Journal Requirements:

“This work was supported by the Dutch Research Council (NWO; https://www.nwo.nl/)

under Grant P16–28 (Project 3). The funders had no role in study design, data

collection and analysis, decision to publish, or preparation of the manuscript”

Reviewers' comments:

Reviewer's Responses to Questions

**Comments to the Author**

1. Is the manuscript technically sound, and do the data support the conclusions?

Reviewer #1: Yes

Reviewer #2: Partly

2. Has the statistical analysis been performed appropriately and rigorously? 

Reviewer #1: Yes

Reviewer #2: No

3. Have the authors made all data underlying the findings in their manuscript fully available?

Reviewer #1: Yes

Reviewer #2: No

4. Is the manuscript presented in an intelligible fashion and written in standard English?

Reviewer #1: Yes

Reviewer #2: Yes

5. Review Comments to the Author

Reviewer #1: The aim of this study was to measure the effectiveness of a training program (changing either cadence or duty factor). There were several other objectives involved, although these were a little confusing. There is merit in the paper, but it could be made a lot simpler so that the main findings were clear and so that practical applications were discussed in more detail. The paper is well written overall with some occasional writing errors. It was unfortunate that the authors did not include line numbering throughout the submission as this made it more difficult to highlight areas for improvement.

Line 9 – I don’t think you need the definite article before “cadence”, in that you can remove “the”. Should you include a citation in this line? You have made other statements in the abstract that could conceivably use a citation so it is not clear why you do this for just the first statement.

Line 32 – please change “may” to “could”. This applies to many other sentences in the manuscript.

Line 38 – please use more formal language and rewrite “lab” as “laboratory”.

Line 41 – note that van Oeveren et al. (2021) state that, “more experienced and faster runners are just as likely, if not more so, to run with lower SF’s instead of higher SF’s.”

Line 45 – when you use the term “in the open field”, do you simply mean “outdoors”?

Line 50 – I think you could come up with a better description of duty factor here. Why not something like “ratio of stance time relative to stride time”, with an earlier explanation that stride time is the time taken for two successive steps? In line 56, it appears that you relate it to a single step time, which is not the same thing.

Line 51 – you do not explain here whether duty factor should be high or low for these factors. For example, it is possible that a low duty factor is beneficial for a lower running economy, but also it is possible that a high duty factor is beneficial for some runners.

Lines 56-61 – it would be useful to know here how you would intend to change cadence or duty factor: would this be by changing stance time, flight time, or both?

Lines 64-76 – so where would particular types of runners lie in this model? For example, you state that the “average” runner would be the ‘sit’ type, but where would we expect sprinters to be, or marathon runners? Women have higher cadences than men, on average, so are they generally ‘hop’ runners? Are any of these running styles better or worse for runners?

Lines 78-95 – the aims are written in quite an unwieldly way. I think you could remove a lot of the discursive parts and state these more clearly. It is quite strange, for example, to make a statement like that in the last sentence – why would you predict that user experience would be relatively good, but that there would be room for improvement? What evidence was this based on?

Line 105 – even with non-parametric statistics, is your sample size still too small for this kind of analysis?

Lines 116-121 – do you have any information about the accuracy or precision of this instrument?

Line 176 – what qualitative methodology was employed in this part of the study?

Line 191 – this appears to be the last time that line numbering is used in the paper, which has made it much harder to provide feedback to you. Please included line numbering for the whole document.

Page 12 – you mention here “marginally” higher values – what is your definition of “marginally”? On the next line, you need to move “only” to after “2” to make it clearer what this sentence means.

Tables – why have you not provided the p-values where these were not below 0.05? It is much better to provide the values rather than simply write “ns”.

Page 14 – you mention at the bottom of this page that “The participants appreciated the scales differently” – could you explain whether this means they interpreted the meaning of the scales differently, or simply rated the apparatus differently?

Page 16 – please change “less changes” to “fewer changes”.

Discussion overall – you need to spend less time repeating the results and more on discussing what they mean. You could try to point out the relevant practical implications at the end of each paragraph.

Page 17 (top part) – it isn’t clear here what the participants are rating; is it the device that was used or the instructions provided?

Page 17 (about line 8) – please include an apostrophe at the end of “participants”.

Page 17 - if there are no population reference values for these “styles”, then how can you allocate people to them? I am not totally sure that using this model helps you in this paper, partly because you have so few participants, but also because it isn’t needed to explain how you can change cadence or duty factor with an audio device. You might want to remove references to the model and make the process simpler for yourself.

Page 17 – please explain to the reader why a ‘bounce’ runner might find it more difficult to decrease duty factor. This is not obvious from your paper.

Page 18 – I don’t know what you mean by “upcoming variables”.

Page 18 – the concluding paragraph is confusing as it isn’t clear whether the point was to see whether the athletes could fit the running styles in the model, or whether it was to see whether the training was beneficial (including from the perspective of using the audio device).

Figures – I think the colored figures are very helpful in seeing the individual responses.

Reviewer #2: The authors aimed at analyzing the effects of modulating cadence and duty factor by using acoustic and verbal instructions. The novelty of the study is basically the overground running condition. The authors used the previously introduced dual- axis model to describe the running style. The aim of the study is plausible. However, the study has some drawbacks: Very low number of participants (1 female, 4 male), non-validated method (?) for stance time detection, lack of effect sizes in the statistics, „sit“ as the preferred running style. My detailed comments are below.

Line 9: It is untypical to cite a paper in abstract.

Line 45: The reference [15] cited and in the following sentence it is written that in these studies possible changes in speed as a result of the cadence manipulation were not specifically analyzed. In the sentence after this sentence it is written that [15] reported an increase in cadence combined with a reduction in speed. Please revise to avoid confusion.

Line 78ff: After the sentence „This study had three aims“, the first two aims are explained with numbering (1-2) and at the end of the introduction the third aim appears. It is hard to follow.

Line 91: It is not clear what you actually mean by the combined and individual instructions. In order to understand, one should first read the whole manuscript. Please revise acordingly.

Line 95: Please define „relatively good“ in terms of your methods.

Line 118: Please provide information or references to prove the validity of the algorithms used within MoveMetrics.

Table 1: The first participant is outlier in terms of the age and the gender. It is understandable that you already have problems with finding participants, however it is not so plausible to write that you searched for 6 months and could only find 5 participants. Maybe to give participants wearable sensors and collect data for 6 months would have also been an alternative.

Line 138-139: „We then calculated the median speed, cadence and duty factor for each instruction, to reduce the effect of possible outliers due to the environment.“ It is not clear how you reduced the effect the possible outliers due to the environment and what you mean by „environment“.

Line 140ff: Please give the effect sizes and consider corrections for multiple comparisons (e.g. Bonferroni-Holm).

6. PLOS authors have the option to publish the peer review history of their article (what does this mean?). If published, this will include your full peer review and any attached files.

Reviewer #1: No

Reviewer #2: No

---

## [Author Response · Author response to Decision Letter 0]

29 Sep 2023

*copied from the uploaded documents*

Point-to-point reply to reviewers’ comments

Reviewer #1: 

The aim of this study was to measure the effectiveness of a training program (changing either cadence or duty factor). There were several other objectives involved, although these were a little confusing. There is merit in the paper, but it could be made a lot simpler so that the main findings were clear and so that practical applications were discussed in more detail. The paper is well written overall with some occasional writing errors. It was unfortunate that the authors did not include line numbering throughout the submission as this made it more difficult to highlight areas for improvement.

Thank you for your review. We apologize for overlooking the fact that the line numbers stopped at the first section break. Line numbering is now applied throughout the document.

Line 9 – I don’t think you need the definite article before “cadence”, in that you can remove “the”. Should you include a citation in this line? You have made other statements in the abstract that could conceivably use a citation so it is not clear why you do this for just the first statement.

We have made the suggested changes.

Line 32 – please change “may” to “could”. This applies to many other sentences in the manuscript.

We have made the suggested change throughout the document

Line 38 – please use more formal language and rewrite “lab” as “laboratory”.

We have made the suggested change throughout the document

Line 41 – note that van Oeveren et al. (2021) state that, “more experienced and faster runners are just as likely, if not more so, to run with lower SF’s instead of higher SF’s.”

We have added more nuance to the statement by adding “, although this is not a consistent finding [22]”.

Line 45 – when you use the term “in the open field”, do you simply mean “outdoors”?

Yes, we have changed the text accordingly.

Line 50 – I think you could come up with a better description of duty factor here. Why not something like “ratio of stance time relative to stride time”, with an earlier explanation that stride time is the time taken for two successive steps? In line 56, it appears that you relate it to a single step time, which is not the same thing.

We agree that the description was unnecessarily complex. We decided not to change it to “ratio of stance time relative to stride time”, because that suggests that the step time of the subsequent step is used in the calculation (stride time = step time(i)+step time(i+1)), which is not the case. Instead we opted to remove “twice the” as this methodological nuance is not relevant at this point in the text. 

Line 51 – you do not explain here whether duty factor should be high or low for these factors. For example, it is possible that a low duty factor is beneficial for a lower running economy, but also it is possible that a high duty factor is beneficial for some runners.

Here we aim to argue why one would want to modify duty factor (similar to the paragraph before for cadence), which may be because it is associated with these factors, but also to further study these associations. We feel that a discussion regarding the exact associations that have been found, especially as these findings are not always conclusive, falls outside of the scope of this paper.

Lines 56-61 – it would be useful to know here how you would intend to change cadence or duty factor: would this be by changing stance time, flight time, or both?

As we intend to change cadence without changing duty factor and the other way around, by definition, both stance time and flight time need to change. In case of a change in cadence without a change in duty factor, both stance time and flight time need to change, but the ratio between them needs to stay the same. In case of a change in duty factor without a change in cadence, the ratio between stance time and flight time needs to change, but the sum of both needs to remain the same. We have added a sentence to include stance time and flight time to give the reader a better idea of the implications.

Lines 64-76 – so where would particular types of runners lie in this model? For example, you state that the “average” runner would be the ‘sit’ type, but where would we expect sprinters to be, or marathon runners? Women have higher cadences than men, on average, so are they generally ‘hop’ runners? Are any of these running styles better or worse for runners?

Those are all very good questions that we would like to know the answers to ourselves. As it stands, the dual-axis model is a theoretical model, for which no reference values are available. More research regarding the model and the running styles defined within it is necessary. The instructions we study in this paper aims to contribute to this research as they provide the potential means to guide runners through the model.

Lines 78-95 – the aims are written in quite an unwieldly way. I think you could remove a lot of the discursive parts and state these more clearly. It is quite strange, for example, to make a statement like that in the last sentence – why would you predict that user experience would be relatively good, but that there would be room for improvement? What evidence was this based on?

We agree with this comment and have removed a lot of the discursive parts and used numbering to clearly distinguish the three aims.

Line 105 – even with non-parametric statistics, is your sample size still too small for this kind of analysis?

We agree that 5 participants is still a limited sample size, which is why we opted for non-parametric testing within each individual participant. We have rephrased the sentence in question to more clearly reflect our method.

Lines 116-121 – do you have any information about the accuracy or precision of this instrument?

Participants used their own sports watches. All participants used a watch from the brand Garmin. Van Oeveren et al. (2019) assessed the accuracy of step frequency measurements of the Garmin Forerunner 620 HRM and found it to be good (ICC, mean±std = 0.85 ±0.06). However, they also state: “since SFs can be measured by foot pods, accelerometers embedded in the watch, or accelerometers embedded in the heart rate monitor, it should be noted that the accuracy may depend on the specific software and hardware setup and thus vary across participants.” We included a paragraph about this topic in the Discussion. 

Line 176 – what qualitative methodology was employed in this part of the study?

We agree, “qualitative” is indeed not the correct term in this context. We have removed it from the text.

Line 191 – this appears to be the last time that line numbering is used in the paper, which has made it much harder to provide feedback to you. Please included line numbering for the whole document.

We apologize for missing the fact that the line numbers stopped at the first section break. Line numbering is now applied throughout the document.

Page 12 – you mention here “marginally” higher values – what is your definition of “marginally”? 

We agree that marginally is not objectively defined. We therefore decided to remove the adjective. 

On the next line, you need to move “only” to after “2” to make it clearer what this sentence means.

We have made the suggested change.

Tables – why have you not provided the p-values where these were not below 0.05? It is much better to provide the values rather than simply write “ns”.

We have added the p-values for the non-significant results, together with the effect sizes.

Page 14 – you mention at the bottom of this page that “The participants appreciated the scales differently” – could you explain whether this means they interpreted the meaning of the scales differently, or simply rated the apparatus differently?

With this sentence we mean that the participants rated the apparatus differently. We understand the confusion and have rewritten this sentence for greater clarity.

Page 16 – please change “less changes” to “fewer changes”.

We have made the suggested change.

Discussion overall – you need to spend less time repeating the results and more on discussing what they mean. You could try to point out the relevant practical implications at the end of each paragraph.

We have rewritten parts of the Discussion in view of your comment.

Page 17 (top part) – it isn’t clear here what the participants are rating; is it the device that was used or the instructions provided?

The participants are rating the training, which consists of the instructions. The device or method that was used, in this case a pre-made audio file, will influence the experience with the training as well, which is why we take this aspect into account in the Discussion. We have rewritten this section to emphasize that they were rating the training.

Page 17 (about line 8) – please include an apostrophe at the end of “participants”.

We have made the suggested change.

Page 17 - if there are no population reference values for these “styles”, then how can you allocate people to them? 

More research is necessary to develop speed-dependent reference values for the dual-axis model. We agree that without existing reference values it did not really make sense to try and guess what the running styles of our participants were. We therefore removed this section.

I am not totally sure that using this model helps you in this paper, partly because you have so few participants, but also because it isn’t needed to explain how you can change cadence or duty factor with an audio device. You might want to remove references to the model and make the process simpler for yourself.

We have long reflected on this comment and see the grounds for removing the model altogether from the paper. However, we have decided not to do so because the dual-axis model forms the conceptual backdrop for this line of research, which has led to a coherent series of publications. Although the dual-axis model as such is not tested in the present study, it provides the rationale for selecting cadence and duty factor as the variables of choice for modulating running style. Moreover, the running styles as discussed in our study are directly derived from the dual-axis model. We therefore respectfully refrained from following your suggestion. 

Page 17 – please explain to the reader why a ‘bounce’ runner might find it more difficult to decrease duty factor. This is not obvious from your paper.

A ’bounce’ runner has a relatively long flight phase. There is a limit to how long the flight phase can be at a certain speed and cadence, as a stance phase is needed as well. When the flight phase is already near this limit to begin with, it would be more difficult to further increase flight phase. We have changed the text to better explain this: ‘However, the preferred running style of a participant is not necessarily the ‘Sit’ running style according to the dual-axis model and hence could influence their ability to change in certain directions, because there is a limit to these variables (e.g. when someone is a pronounced ‘Bounce’ runner (a relatively long flight phase), it could be more difficult to decrease duty factor even further, as the relative flight phase cannot be increased indefinitely).’

Page 18 – I don’t know what you mean by “upcoming variables”.

We have changed it to: ‘stance time and duty factor have only recently been recognized as important running variables’.

Page 18 – the concluding paragraph is confusing as it isn’t clear whether the point was to see whether the athletes could fit the running styles in the model, or whether it was to see whether the training was beneficial (including from the perspective of using the audio device).

After rewriting the aims in the Introduction section, we made small changes in the concluding paragraph of the Discussion to better align the conclusions with the aims.

Figures – I think the colored figures are very helpful in seeing the individual responses.

Thank you

 

Reviewer #2: 

The authors aimed at analyzing the effects of modulating cadence and duty factor by using acoustic and verbal instructions. The novelty of the study is basically the overground running condition. The authors used the previously introduced dual- axis model to describe the running style. The aim of the study is plausible. However, the study has some drawbacks: Very low number of participants (1 female, 4 male), non-validated method (?) for stance time detection, lack of effect sizes in the statistics, „sit“ as the preferred running style. My detailed comments are below.

Thank you for your review. We are aware that the study has some limitations. We have highlighted those in the Discussion. Nevertheless, our results are sufficient to support the paper’s conclusion that running style can be modulated in overground running by combining acoustic pacing and verbal instructions, which is a finding worthy of sharing with the research community. Please find below our point-to-point reply to your comments. 

Line 9: It is untypical to cite a paper in abstract.

We have removed the citation from the abstract

Line 45: The reference [15] cited and in the following sentence it is written that in these studies possible changes in speed as a result of the cadence manipulation were not specifically analyzed. In the sentence after this sentence it is written that [15] reported an increase in cadence combined with a reduction in speed. Please revise to avoid confusion.

We have added the word ‘often’ to better reflect that most studies did not analyze an effect on speed, but, indeed, one study did.

Line 78ff: After the sentence „This study had three aims“, the first two aims are explained with numbering (1-2) and at the end of the introduction the third aim appears. It is hard to follow.

We have rewritten this section by removing unnecessary text, and used numbering to clearly distinguish the three aims.

Line 91: It is not clear what you actually mean by the combined and individual instructions. In order to understand, one should first read the whole manuscript. Please revise accordingly.

We have added a short explanation in the Introduction to clarify this point (lines 79-82).

Line 95: Please define „relatively good“ in terms of your methods.

We decided to remove this sentence, because reviewer 1 pointed out that the statement was not based on any evidence.

Line 118: Please provide information or references to prove the validity of the algorithms used within Move-Metrics.

Move-Metrics only calculated descriptive statistics based on the uploaded data. We feel that it is safe to assume that the calculations of the mean, standard deviation, median, inter quartile range (IQR), minimum, maximum, and the lower (5%) and upper (95%) limit of the confidence interval are valid. 

Because the participants used their own sports watches, it is useful to say something about the validity and reliability of these watches as measurement devices. We have added a section to the Discussion addressing this aspect.

Table 1: The first participant is outlier in terms of the age and the gender.

Since all the statistic comparisons are within each participant, this should not have impacted the conclusions.

 It is understandable that you already have problems with finding participants, however it is not so plausible to write that you searched for 6 months and could only find 5 participants. Maybe to give participants wearable sensors and collect data for 6 months would have also been an alternative.

Before starting recruitment we also thought it wasn’t plausible that only 5 participants would be willing and eligible to participate. The main problem that we ran into was the fact that the devices of most people who wanted to participate did not measure stance time. When trying to recruit people who did have devices that measured stance time, almost all of them were training for a specific event, such as a marathon, and were not willing to train with the audio file for 8 weeks and thereby deviate from their own training regime. 

Giving participants wearable sensors for 6 months was actually the first option we explored, but this was not feasible within our financial budget (as we set out to recruit a lot more than 5 participants). If the recruitment had gone as we expected (based on van Oeveren et al. 2019), we would be presenting the results of at least 50 participants. 

While we agree that the sample is very limited, the results demonstrate that running style can be modulated in overground running by combining acoustic pacing and verbal instructions and provide insight into both intended and covarying effects of the tested instructions.

Line 138-139: „We then calculated the median speed, cadence and duty factor for each instruction, to reduce the effect of possible outliers due to the environment.“ It is not clear how you reduced the effect the possible outliers due to the environment and what you mean by „environment“.

In general, the median is more robust and less influenced by outliers than for example the mean. By ‘environment’, we literally mean the environment they ran in. Because the runners ran in their own neighborhoods, we do not know what the environment was like. If a run was more uphill, this would have affected speed, cadence and stance time. 

We have removed the reasoning from the sentence to avoid confusion, as it is not relevant to the paper and the small sample size already indicates the use of the median rather than the mean.

Line 140ff: Please give the effect sizes and consider corrections for multiple comparisons (e.g. Bonferroni-Holm).

We have added the effect sizes to the tables. However, as we do not make multiple comparisons on the same data, we did not apply Bonferroni corrections.

---

## [Decision Letter · Decision Letter 1]

22 Nov 2023

Exploring running styles in the field through cadence and duty factor modulation

PONE-D-23-18858R1

Dear Dr. Nijs,

We’re pleased to inform you that your manuscript has been judged scientifically suitable for publication and will be formally accepted for publication once it meets all outstanding technical requirements.

Kind regards,

Laurent Mourot

Section Editor

PLOS ONE

Additional Editor Comments (optional):

Reviewers' comments:

Reviewer's Responses to Questions

**Comments to the Author**

1. If the authors have adequately addressed your comments raised in a previous round of review and you feel that this manuscript is now acceptable for publication, you may indicate that here to bypass the “Comments to the Author” section, enter your conflict of interest statement in the “Confidential to Editor” section, and submit your "Accept" recommendation.

Reviewer #1: All comments have been addressed

Reviewer #2: All comments have been addressed

2. Is the manuscript technically sound, and do the data support the conclusions?

Reviewer #1: Yes

Reviewer #2: Yes

3. Has the statistical analysis been performed appropriately and rigorously? 

Reviewer #1: Yes

Reviewer #2: Yes

4. Have the authors made all data underlying the findings in their manuscript fully available?

Reviewer #1: Yes

Reviewer #2: Yes

5. Is the manuscript presented in an intelligible fashion and written in standard English?

Reviewer #1: Yes

Reviewer #2: Yes

6. Review Comments to the Author

Reviewer #1: Thank you for making the recommended changes. Your paper reads much better now and has some interesting findings for practitioners.

Reviewer #2: Thank you for addressing all my comments.

Please make sure you are consistent in british/american english (e.g. analysed/analyzed) before your final submission.

7. PLOS authors have the option to publish the peer review history of their article (what does this mean?). If published, this will include your full peer review and any attached files.

Reviewer #1: No

Reviewer #2: No

---

## [Editor Report · Acceptance letter]

27 Nov 2023

PONE-D-23-18858R1 

Exploring running styles in the field through cadence and duty factor modulation 

Dear Dr. Nijs:

I'm pleased to inform you that your manuscript has been deemed suitable for publication in PLOS ONE. Congratulations! Your manuscript is now with our production department. 

Kind regards, 

on behalf of

Dr Laurent Mourot 

Section Editor

PLOS ONE